# Exploring Codon Adjustment Strategies towards *Escherichia coli*-Based Production of Viral Proteins Encoded by HTH1, a Novel Prophage of the Marine Bacterium *Hypnocyclicus thermotrophus*

**DOI:** 10.3390/v13071215

**Published:** 2021-06-23

**Authors:** Hasan Arsın, Andrius Jasilionis, Håkon Dahle, Ruth-Anne Sandaa, Runar Stokke, Eva Nordberg Karlsson, Ida Helene Steen

**Affiliations:** 1Department of Biological Sciences, University of Bergen, N-5020 Bergen, Norway; Ruth.Sandaa@uib.no (R.-A.S.); Runar.Stokke@uib.no (R.S.); 2Centre for Deep Sea Research, University of Bergen, N-5020 Bergen, Norway; Hakon.Dahle@uib.no; 3Division of Biotechnology, Lund University, P.O. Box 124, SE-221 00 Lund, Sweden; Andrius.jasilionis@biotek.lu.se (A.J.); Eva.nordberg_karlsson@biotek.lu.se (E.N.K.); 4Computational Biology Unit, University of Bergen, N-5020 Bergen, Norway

**Keywords:** prophage, hydrothermal vent, *Hypnocyclicus thermotrophus*, lytic cassette, *Escherichia coli*, heterologous expression, codon optimization, codon harmonization

## Abstract

Marine viral sequence space is immense and presents a promising resource for the discovery of new enzymes interesting for research and biotechnology. However, bottlenecks in the functional annotation of viral genes and soluble heterologous production of proteins hinder access to downstream characterization, subsequently impeding the discovery process. While commonly utilized for the heterologous expression of prokaryotic genes, codon adjustment approaches have not been fully explored for viral genes. Herein, the sequence-based identification of a putative prophage is reported from within the genome of *Hypnocyclicus thermotrophus*, a Gram-negative, moderately thermophilic bacterium isolated from the Seven Sisters hydrothermal vent field. A prophage-associated gene cluster, consisting of 46 protein coding genes, was identified and given the proposed name *Hypnocyclicus thermotrophus* phage H1 (HTH1). HTH1 was taxonomically assigned to the viral family *Siphoviridae*, by lowest common ancestor analysis of its genome and phylogeny analyses based on proteins predicted as holin and DNA polymerase. The gene neighbourhood around the HTH1 lytic cassette was found most similar to viruses infecting Gram-positive bacteria. In the HTH1 lytic cassette, an N-acetylmuramoyl-L-alanine amidase (Amidase_2) with a peptidoglycan binding motif (LysM) was identified. A total of nine genes coding for enzymes putatively related to lysis, nucleic acid modification and of unknown function were subjected to heterologous expression in *Escherichia coli*. Codon optimization and codon harmonization approaches were applied in parallel to compare their effects on produced proteins. Comparison of protein yields and thermostability demonstrated that codon optimization yielded higher levels of soluble protein, but codon harmonization led to proteins with higher thermostability, implying a higher folding quality. Altogether, our study suggests that both codon optimization and codon harmonization are valuable approaches for successful heterologous expression of viral genes in *E. coli*, but codon harmonization may be preferable in obtaining recombinant viral proteins of higher folding quality.

## 1. Introduction

Hydrothermal vents host some of the most diverse microbial communities in marine environments. Diverse (hyper)thermophilic bacteria and archaea grow within the steep chemical and temperature gradients formed by rapid mixing of high temperature (up to above 300 °C) reduced vent fluids and cold seawater [1,2]. The discovery of the hydrothermal vent ecosystem remains one of the biggest breakthroughs in our understanding of how life can be sustained in extreme conditions, marked by the first vent observation on the Galápagos Rift, in the eastern Pacific [3] and the discovery of the first black smoker vents [4]. Today, hydrothermal vents are well-known as attractive sites for bioprospecting of biotechnologically interesting enzymes [5,6,7,8] and other valuable biomolecules with potential industrial applications [6,9,10]. As with other marine biomes [11,12,13], hydrothermal vent environments are observed to be abundant with viruses, especially tailed dsDNA bacteriophages of order *Caudovirales* [14,15]. These viruses remain a largely unexplored space of genetic diversity and, therefore, an under-utilized source for enzyme bioprospecting efforts [16,17].

The unique biology of host-reliant viral replication makes viruses remarkably interesting entities for biotechnology, where lytic enzymes can be found associated with their strategy of host infection [18,19]. While lytic phages reproduce by host cell lysis, lysogenic or temperate phages can remain dormant until induction, either as so-called “prophages” integrated into the host genome, or as extrachromosomal elements [20,21]. Temperate phages have been reported as particularly present in the microbial communities associated with vent fields [15,22], likely related to challenging environmental factors such as lower host abundances, limiting nutrient availability, and the fringe physical and chemical conditions present at these sites. In addition, the set of viral genes made available to the host via lysogeny may also produce fitness-enhancing phenotypes, increasing the host resilience in these environments [23,24,25].

The currently studied minority of bacteriophages have yielded numerous biotechnologically important enzymes. Some significant examples include enzymes acting on nucleic acids, such as DNA polymerases, DNA ligases from bacteriophages T4 [26,27] and T7 [28,29,30], and exonuclease from the bacteriophage T5 [31]. Furthermore, lytic enzymes such as endolysins, naturally arming the phages for the degradation of bacterial cell walls, are of increasing interest as bactericidal agents [32,33,34,35] and have been subjected to trials as phage therapy [36,37]. Many of the above viruses were studied from isolates and provide a glimpse into similar discoveries possible from within the vast viral sequence space in marine environments [13,16].

To be able to study discovered viral enzymes of potential biotechnological interest, molecular cloning and heterologous expression approaches are required to produce the enzymes in amounts needed for characterization experiments. Study of the heterologous expression of viral genes from marine metagenomes, however, has been extremely limited [38]. Extending the knowledge in this field has subsequently been a major task in the project Virus-X (Viral Metagenomics for Innovation Value) aiming to identify and characterize novel enzymes and other proteins from bacteriophages and archaeal viruses. To date, only a few examples of studies describing the expression of viral genes from environmental marine resources are reported [39,40]. For the heterologous production of most proteins, *Escherichia coli* remains a desirable host due to its ease of use, quick generation times and a wide genetic toolkit regarding cloning and expression vectors [41]. However, *E. coli* does present certain well-documented challenges in soluble protein production when expressing genes from genetically less-related sources [41,42]. Furthermore, the distinct codon usage bias of *E. coli* often presents a difference in the availability of tRNAs between the native organism and itself, adversely affecting protein expression efficacy [43,44].

Numerous approaches exist to increase soluble protein yields of recombinant genes in *E. coli*. The use of various fusion protein tags has been a popular and effective way to improve soluble yields for many years [45,46,47,48]. The use of transcription-level adjustments to improve soluble protein expression has been described in recent years, initially as “codon optimization” [49] and later as “codon harmonization” [50]. Both of these approaches rely on the modification of codons in the DNA sequence of the target prior to expression, to code for the same eventual polypeptide, but with a set of tRNAs tailored for the machinery of the expression host. The difference among these approaches can be summarized as such: codon optimization substitutes rare codons in the native gene sequence with those that are most abundant in the heterologous host, potentially allowing a high-speed protein production, whereas codon harmonization aims to replicate the cadence of native gene expression in the host, potentially allowing for correct protein folding during expression. While codon optimization has been widely demonstrated to have some degree of success in expressing genes from a diverse range of native hosts [48,51], including viruses [52,53,54], codon harmonization is a more recent approach and, to our knowledge, has not yet been explored towards the expression of viral genes in *E. coli*.

In this work, we report the first study of a temperate phage infecting *H. thermotrophus:* a free-living, Gram-negative, moderately thermophilic bacterium isolated from a microbial mat collected from the Seven Sisters hydrothermal vent field located on the Arctic Mid-Ocean Ridge [55,56]. Within the phylum *Fusobacteria, Hypnocyclicus thermotrophus* IR-2^T^ (=DSM 100055 =JCM 30901) is listed as the current type strain of the genus *Hypnocyclicus*. In addition to describing the identification, gene organization and taxonomic analysis of the prophage via in silico methods, we also report on our efforts to identify and recombinantly express genes with potential links to various lytic and nucleic acid modifying enzymatic activities. In an effort to facilitate the soluble heterologous production of proteins in *E. coli,* we implemented the codon optimization and harmonization approaches in parallel for a set of nine diverse enzyme candidates. The comparison of proteins produced via these approaches revealed notable differences in their soluble yields and thermostability. Altogether, the combined strategy used herein presents a cohesive application of both bioinformatics and molecular biology to improve access to the viral genetic diversity present in marine environments.

## 2. Materials and Methods

### 2.1. Identification and Annotation of Prophage Genes

The annotated genome assembly of the bacterium *H. thermotrophus* was downloaded from NCBI GenBank (RefSeq GCF_004365575.1). Manual analysis of the genome indicated presence of prophage genes. To further assess these putative prophage genes, the GenBank file of the assembly was uploaded to the PHASTER (https://phaster.ca/, accessed on 10 December 2019) [57,58] online tool and compared against the PHASTER prophage/virus database (last updated in August 2019). The analysis output described the genome region(s) containing the prophage genes, along with putative functional annotations. In addition to annotations provided by NCBI and PHASTER, the HHpred server (https://toolkit.tuebingen.mpg.de/tools/hhpred, accessed on 1 December 2020) [59,60,61], and the eggNOG-Mapper (http://eggnog-mapper.embl.de accessed on 12 December 2019) [62,63] online services were also used for the functional annotation of the prophage genes using corresponding amino acid sequences.

When using the HHpred server for the pairwise comparison of profile hidden Markov models (HMMs), the databases queried were PDB_mmCIF70_29_Nov, Pfam-A_v33.1, COG_KOG_v1.0 and NCBI_Conserved_Domains(CDs)_v3.18.

### 2.2. Taxonomic Analysis of HTH1

To taxonomically characterize HTH1, the genes identified as phage-related using the PHASTER tool were subjected to a translated nucleotide to protein BLAST (blastx, accessed on 2 April 2020) search. The following parameters: organism = viruses (txid:10239), number of alignments = 100, word size = 6 were used. The resulting hits were then parsed and taxonomically assigned by lowest common ancestor (LCA) analysis [64] in MEGAN software (version 6.18.6) (Tübingen, Germany) [65]. The following parameters were used: minimum support = 2, minimum score = 70, top percent = 10. Megan Mapping Database file version October 2019 was used.

With a reported success rate of 93% when assigning tailed and unclassified phages to their defined head–neck–tail-based categories, the “Remote Homology Detection of Viral Protein Families—Virfam” [66] (http://biodev.cea.fr/virfam, accessed on 3 April 2020) server was also used to further analyse the taxonomy of HTH1.

### 2.3. Analysis of Prophage Host Range

In order to analyse the currently documented host range of similar phages, DNA sequence of HTH1 was used to perform a translated nucleotide–protein BLAST (blastx) search as described above, except using the NCBI non-redundant (nr) nucleotide database. The species names of the top 5000 hits were parsed and uploaded to phyloT (https://phylot.biobyte.de/, accessed on 5 April 2020) (version 2) [67] online tool to visualize the taxonomic distribution by generating a phylogeny of the cumulative NCBI taxonomy lineages of each species on the list (Appendix A).

### 2.4. Phylogeny Analyses

The amino acid sequence of the holin (GenBank WP_134112787.1) identified in HTH1 was used as a basis for phylogeny analyses and relationship of the prophage to viruses in the NCBI (nr) database. A protein–protein BLAST (blastp) search was performed via NCBI BLAST [68] with the following parameters: organism = viruses (txid:10239), word size = 6. A list of 94 proteins exported from the BLAST search (including the holin from HTH1) was aligned using MAFFT (version 7.453) [69]. Gap regions were trimmed with trimAl (version 1.2 rev59) [70] using the ‘*gappyout*’ command to automatically trim sequences based on gaps in the alignment. The resulting trimmed alignment comprising 106 amino acid positions was manually analysed and used as a basis to infer maximum likelihood (ML) phylogeny using IQ-TREE (version 1.6.12) [71] tool. The best-fitting model was automatically determined by ModelFinder [72], and ultrafast bootstrapping was performed with 1000 replicates [73]. The best-fitting model was identified as LG+I+G4 (general matrix with invariable site plus discrete gamma model [74,75]). The resulting tree was then annotated using the online Interactive Tree of Life (iTOL) (https://itol.embl.de/, accessed on 5 April 2020) (version 5.5.1) [76] software. Branches with less than 50% bootstrap support were collapsed (Figure 1).

The amino acid sequence of HTH1 holin was further analysed using a protein–protein BLAST (blastp) against the Integrated Microbial Genomics/Virus (IMG/VR) (https://img.jgi.doe.gov/vr/, accessed on 10 April 2020) [77] and Ocean Gene Atlas (http://tara-oceans.mio.osupytheas.fr/ocean-gene-atlas/, accessed on 10 April 2020) [78] databases to compare the prophage to viral genes from environmental samples, metagenomic datasets and other non-isolated virus genes. The top 100 hits with the highest percent identity from the IMG/VR search and all the hits (18) from the Ocean Gene Atlas were extracted in addition to the 94 sequences from NCBI as described above. After automatic and manual curation to remove duplicates or non-holin hits, a total of 211 holin-related sequences were aligned, trimmed and visualized as described above, with the best-fitting ML model for this group of sequences identified as LG+F+I+G4 (general matrix with invariable site plus discrete gamma model [74,75] with empirical codon frequencies counted from the data) (Appendix A).

The putative DNA polymerase (HTP4385) (GenBank WP_134112782.1) was also subjected to phylogeny analysis, using the same parameters as described above for the holin-based tree. In this analysis, a list of 101 protein entries was used to create a 625 amino acid long alignment for the construction of the tree shown in Appendix A.

### 2.5. Gene Neighbourhood Analysis

Gene neighbourhoods between genes of HTH1 and three highly similar viral gene clusters was compared. The similar viral gene clusters were selected based on the closest alignments to the HTH1 holin in the extended tree shown in Appendix A. Alongside HTH1, marine anoxygenic phototropic community R3 (MAPCR3) (IMG scaffold ID: Ga0071011_100294), *Streptococcus* phage Javan630 (SPJ630) (NCBI:txid2548289) and the *Erysipelothrix* phage phi1605 (EP1605) (NCBI:txid2006938) were inspected using GeneGraphics (https://katlabs.cc/genegraphics/app, accessed on 20 April 2020) [79] (Figure 2) by uploading the relevant genome regions with their annotations for each entry in NCBI GenBank format to the online tool.

### 2.6. Selection of Genes for Expression Trials

In addition to the genes constituting the lytic cassette, genes with various putative functions on either side of the HTH1 lytic cassette were analysed. After inspection, nine genes were selected for expression trials for their putative activities related to lysis and DNA replication, including three genes with hypothetical function or conserved domains of unknown function (DUF). The selected genes were labelled with the prefix HTP (*H. thermotrophus* phage) followed by the last four digits of their corresponding locus tag in the NCBI GenBank annotation (such as HTP4435). The selected genes and their annotated domain structures predicted by the HMMER web service (https://www.ebi.ac.uk/Tools/hmmer/search/phmmer, accessed on 20 April 2020) [80,81,82] were visualized in Figure 3.

### 2.7. Preparation of Sequences for Protein Expression of Selected Genes

Codon optimization [49] and codon harmonization approaches [83,84] were used in parallel to evaluate their effectivity in obtaining properly folded, soluble protein from each of the selected genes tailored for heterologous expression in *E. coli*. Codon-optimized gene sequences were generated via GenSmart Codon Optimization (GenScript, Piscataway, NJ, USA) online tool following default codon optimization parameters. Codon Harmonizer developed by Claassens et al. [51] online tool was used to harmonize codon usage frequencies between the prophage host *H. thermotrophus* NCBI GenBank (RefSeq GCF_004365575.1) and the heterologous expression host *E. coli* BL21(DE3) NCBI GenBank (GenBank GCA_000022665.2) (accessed in April 2019). Codon Adaptation Index (CAI) and Codon Harmonization Index (CHI) values were calculated for each sequence. Both the codon-optimized and codon-harmonized target protein gene sequences (Appendix A) were ordered to be synthesized and delivered pre-cloned in pET-21b(+) (Merck, Darmstadt, Germany) [85] vector (GenScript, Leiden, the Netherlands), featuring a C-terminal hexa-histidine tag [86] to facilitate purification using affinity chromatography.

### 2.8. Protein Production in E. coli

All expression constructs were transformed into *E. coli* BL21(DE3) (Merck, Darmstadt, Germany) cells using the heat-shock protocol provided by the manufacturer, using 30 ng of plasmid per 15 µL of bacteria suspension. Single colonies were picked from Lysogeny Broth (LB)-agar plates containing 100 µg/mL ampicillin after plating and overnight growth at 37 °C, and 10 mL pre-cultures in LB were subsequently inoculated and incubated overnight at 37 °C with 220 rpm shaking. Expression cultures in Tryptic Soy Broth (Merck, Darmstadt, Germany) (adjusted to pH 7.4/RT) at 100 mL scale were inoculated with 5% (*v/v*) of each pre-culture and were grown at 37 °C and 220 rpm until an optical density at 600 nm of 0.5–0.6 was reached. The incubation temperature was then reduced to 28 °C and allowed to equilibrate for 30 min. Expression was induced with 0.5 mM isopropyl β-D-1-thiogalactopyranoside, at 28 °C for 5 h. Following the expression, cells were harvested by centrifugation at 5000× *g* at 4 °C for 10 min. Collected cells were re-suspended in 10 mL of lysis buffer containing 50 mM Tris-HCl pH 7.4/RT, 60 mM imidazole, 500 mM NaCl and 5% (*v/v*) glycerol and were lysed using ultrasonication performed at 4 °C using 5 × 30 s bursts at 15 s intervals, with 25% amplitude. An aliquot representing the total protein fraction was taken and stored at 4 °C from each crude lysate before clarification of lysates by centrifugation at 12,000× *g* at 4 °C for 3 min. After clarification, aliquots were taken from all samples representing the soluble protein fraction and stored at 4 °C.

### 2.9. Protein Solubility Assessment and Yield Estimation

Aliquots taken from lysed cell pellets, representing the total protein (crude lysate) and soluble protein (clear lysate) fractions were run on a gradient (8–16%) SDS-PAGE gel (GenScript, Piscataway, NJ, USA) to assess expression levels. Precision Plus Dual Color (Bio-Rad, Hercules, CA, USA) protein ladder was used for protein molecular mass determination. Equivalent volumes of protein samples were loaded onto the electrophoresis gels seeking to fractionate equal protein amounts. The gel was run at 200 V, and subsequently stained using InstantBlue (Expedeon, Cambridge, UK) using a staining protocol provided by the manufacturer. After staining was complete, unbound dye was washed off the gel using distilled water on a benchtop shaker to reveal protein bands. The gels were photographed using MiniBIS Pro system processing images with GelCapture (version 7.0.15) suite (DNR Bio-Imaging Systems, Neve Yamin, Israel).

Densitometry calculations to determine relative abundance of target proteins in the soluble lysate fractions were performed using GelQuantum Pro (version 12.2) suite (DNR Bio-Imaging Systems, Neve Yamin, Israel). Total protein concentration was measured with a NanoDrop 1000 spectrophotometer (operating software version 3.7; Thermo Fisher Scientific, Waltham, MA, USA), assuming A_280_ 1 = 1 mg/mL. Target protein soluble yields were estimated by combining the results of densitometry and total soluble protein quantification.

### 2.10. Protein Purification

HTH1 proteins obtained in soluble form were purified to near homogeneity from clear lysate fractions by nickel affinity chromatography. Soluble protein fraction in lysis buffer was loaded 1 mL/min into a HisTrap HP 1 mL (7 mm × 25 mm) column (Cytiva, Uppsala, Sweden) equilibrated with lysis buffer. Target proteins were eluted (2 column volumes (CV)) with elution buffer containing 50 mM Tris-HCl pH 7.4/RT, 500 mM imidazole, 500 mM NaCl and 5% (*v/v*) glycerol at 1 mL/min after extensive washing (5–8 CV) of unbound proteins with lysis buffer. The purified proteins were stored in elution buffer at 4 °C after filtering twice through regenerated cellulose 0.2 μm pore size syringe filters (GE Healthcare, Uppsala, Sweden).

Protein integrity and purity were assessed via SDS-PAGE. Protein concentrations were measured spectrophotometrically, considering calculated absorption coefficients for pure proteins. Purification yields were calculated comparing the target protein amount in the soluble protein fractions with the target protein amount obtained after the purification and filtration steps.

### 2.11. Protein Thermal Unfolding Assay

Nanoscale differential scanning fluorometry based on internal tryptophane as well as tyrosine content was performed to determine the melting temperatures (T_m_, °C) of purified HTH1 proteins. These measurements were carried out on a Prometheus NT.48 system using standard grade capillaries (NanoTemper Technologies, Munich, Germany). The purified protein samples were diafiltrated into assay buffer containing 50 mM Tris-HCl pH 7.4/RT and 2% (*v/v*) glycerol using Amicon Ultra-0.5 mL (3 Kda) centrifugal filters (Merck, Darmstadt, Germany). Protein concentrations were adjusted to 0.2 mg/mL with assay buffer after diafiltration. Thermal unfolding assays were performed at adjusted 40% excitation power, with a temperature gradient between 20–95 °C and at a ramp rate of 1 °C/min. Finally, analysis of the recorded emission intensities, emission ratio (350 nm/330 nm) and first derivative calculations were processed using the PR.ThermControl software (version 2.0.4) (NanoTemper Technologies, Munich, Germany).

## 3. Results

### 3.1. Functional Annotation and Taxonomy Analysis of HTH1

Three regions of putative viral origin were identified within the *H. thermotrophus* using the PHASTER tool [58]. Region 1 (Appendix A) was reported as an incomplete prophage region (PHASTER score: 10), consisting of eight conserved domains (CDs) from locus tags EV215_RS03310 to EV215_RS03345 in the sense (+) strand. Region 2 was also predicted as incomplete (PHASTER score: 50), consisting of 33 CDs from locus tags EV215_RS04355 to EV215_RS04515. However, attachment sites attL and attR (nucleotide sequence TTACCATCTTA) were found between locus tags EV215_RS04470-EV215_RS04475 and EV215_RS04435-EV215_RS04440, respectively, within region 2, indicating that this region was likely associated with viral interaction and virus integration on to the host genome. Region 3 was predicted to be an intact prophage region (PHASTER score: 100) and contained 29 CDs from locus tags EV215_RS04440 to EV215_RS04580. There was an 11,971 bp overlap between regions 2 and 3, representing 16 CDs, with both regions found on the complementary (−) strand of the genome. Furthermore, regions 2 and 3 showed highly similar average G + C contents, 37.7% and 38.5%, respectively. In comparison, the average G + C contents of region 1 and the host genome were 27.9% and 24.8%, respectively. Due to their overlap, and coherent composition, regions 2 and 3 were considered as the “complete” prophage genome, totalling 46 CDs and a genome size of 41,571 bp. This region was subsequently designated with the proposed name *Hypnocyclicus thermotrophus* phage H1 (HTH1). With the combined use of various pipelines, functional annotations could be suggested for 34 HTH1 genes. The remaining 12 were noted as hypothetical, or to contain unknown elements as listed in Appendix A.

Taxonomic analysis based on the LCA algorithm in MEGAN suggested affiliation of HTH1 with the family *Siphoviridae* and the order *Caudovirales*. Consistently, the Virfam analysis (resulting identities provided in Appendix A) identified the prophage head–neck–tail modules as being part of “Neck Type 1—Cluster 2” type of phages, noted to be associated with siphoviruses. Holin genes have previously been suggested as a phage-specific signature gene for siphoviruses [87]. Phylogeny analyses based on the HTH1 holin (Figure 1) as well as DNA polymerase (Appendix A) amino acid sequences revealed the closest affiliations to known phages from the Javan group of *Streptococci* phages [88] and to the *Erysipelothrix* phage phi1605 (NCBI:txid2006938). The sequence identity between the HTH1 holin and the holins from *Streptococcus* phage Javan630 (SPJ630) and *Erysipelothrix* phage phi1605 (EP1605) was found to be 75.7% and 75.0%, respectively.

The closest identified holin homologue from another phage infecting Gram-negative bacteria was that of the phage Funu2 (NCBI:txid1640978) (Figure 1), which is reported to infect *Fusobacterium nucleatum* [89] (sequence identity of 38.6%). This is an interesting hit, as to date, studies of viruses and viral genes associated with *Fusobacteria* remain limited, with only a small number of phages characterized thus far [37,90,91,92].

When the HTH1 holin was compared against environmental sequences from IMG/VR, an even closer hit at 99% sequence identity was observed against a metagenome-derived holin from a marine anoxygenic phototrophic community R3 (MAPCR3) sample (IMG genome ID 3300004816) originating from a shallow salt marsh pool in Falmouth, MA, USA (Appendix A). When the gene neighbourhood surrounding the lytic cassette of HTH1 was compared with those of MAPCR3, SPJ630 and the EP1605 (Figure 2), a remarkably close similarity was identified between the HTH1 and MAPCR3 lytic cassettes, particularly over the four genes corresponding to HTP4425 to HTP4410 in HTH1 (Appendix A). The similarity was less significant when comparing to cassettes of SPJ630 and EP1605. Furthermore, the lytic cassette amidase (HTP4410) was observed to be replaced by a second glycosyl hydrolase (CAZy GH25) in SPJ630 and EP1605 when the gene annotation and protein domain structures were reviewed using a HMMER search [82] (Figure 3).

### 3.2. Selection of Genes for Expression Trials

HTH1 genes with annotations related to roles in lysis and DNA replication were examined further, examining protein domain structures through comparisons to multiple sequence databases (Appendix A). A set of nine genes were chosen for protein expression trials, as shown in Figure 3, with their designations and associated domain structures. Gene targets associated with the prophage lytic cassette (defined in Section 3.1), including holin (HTP4415), glycosyl hydrolase (HTP4420) and the amidase with a LysM domain (HTP4410), were selected for their putative role in cell lysis, in addition to the phage tail protein (HTP4435) with associations to endopeptidase activity. The hypothetical gene HTP4425 neighbouring the glycosyl hydrolase (HTP4420) was also picked for its potential connection to the lysis-related cluster. Two genes annotated with nucleotide cleavage and production activities were also selected: the rRNA biogenesis protein RRP5 (HTP4400) with putative endonucleolytic activity towards rRNA, and the DNA polymerase I (HTP4385). Furthermore, two genes flanking the HNH endonuclease, HTP4360 and HTP4350, were picked for their potential associations with nucleolytic activity. The gene HTP4350 (GenBank: WP_134112775.1) was annotated as “DUF262 domain containing protein” by the NCBI pipeline; however, a putative DNase activity was also suggested when analysed with HHpred (Appendix A), and it is upstream of the prophage gene region in the *H*. *thermotrophus* genome.

Searches made against PDB for structural insight pertaining to the nine HTH1 proteins revealed only low similarity hits for three proteins, HTP4420, HTP4410 and HTP4350, to PDB entries 4S3J, 3HMB and 1D9D, respectively (Appendix A). However, all three structures reported associations with the expected functions in the HTH1 proteins, such as peptidoglycan lysis for HTP4410 and HTP4350, and DNA polymerase for HTP4350 (Appendix A).

### 3.3. Expression of Target Codon-Adjusted Gene Variants

The codon frequencies of the HTH1 gene sequences were analysed, estimating CAI for the native host *H*. *thermotrophus*. All target protein genes demonstrated CAI values of approximately 0.4–0.5 (Table 1). Estimated CAI values indicated that HTH1 gene sequences were moderately adapted for expression in the native host, predicting comparatively moderate native expression level of the target proteins. Target genes were subsequently processed to generate codon-optimized and codon-harmonized gene sequence variants, adjusted from the *H*. *thermotrophus* codon usage bias towards compatibility with the expression host *E*. *coli* BL21(DE3). Quantitative analysis of codon-adjusted sequence variants confirmed the expected levels of codon adaptation (Table 1). The CAI of codon-optimized gene sequences varied between 0.84 and 0.89, indicating high adaptation towards heterologous expression in *E*. *coli*. Codon-harmonized sequences, as expected, were less adapted to be expressed in the selected strain, with CAI varying between 0.58 and 0.74. It was noted that CAI of codon-harmonized sequences showed higher variation compared to CAI of codon-optimized sequences. The CHI values of codon-optimized variants were 0.12–0.13 below (Table 1) the estimated CHI values from codon-harmonized sequences, confirming an expected trend for more substantial changes imposed on codon-optimized variants. Moreover, the CHI value of each codon-harmonized gene variant was similar and between 0.43 and 0.48. Even though CHI comparison indicated that codon-harmonized variants were closer to native codon sequences of target protein genes, the “harmonization” effect observed could be interpreted as moderate [51].

All nine codon-optimized gene variants were successfully expressed in *E*. *coli,* at different levels (data not shown). However, the hypothetical protein (HTP4425), glycosyl hydrolase (HTP4420), holin (HTP4415) and the DNA polymerase I (HTP4385) were not detected in the soluble protein fraction, as estimated by SDS-PAGE. Insolubility was particularly expected for the holin because of the multiple transmembrane helices present in the structure (Figure 3), and no significant difference was observed from the use of either codon adjustment approach. Among the codon-harmonized set of genes, expression in *E. coli* could not be observed for the genes encoding the holin (HTP4415) as well as the hypothetical protein (HTP4360). For the other seven genes, only four were found to yield soluble proteins. These proteins were the endopeptidase tail protein (HTP4435), amidase (HTP4410), rRNA biogenesis protein RRP5 (HTP4400) and DUF262 / DNase (HTP4350) (Table 1).

In total, implementation of codon adjustment approaches for selected HTH1 genes resulted in the soluble protein production from five codon-optimized and four codon-harmonized gene variants (Figure 4). The set of soluble proteins expressed from codon-optimized and codon-harmonized variants differed by the hypothetical protein (HTP4360) that was not found expressed as soluble from its codon-harmonized variant. As typically expected [50,93], expression levels estimated by densitometry analyses for the five common soluble protein targets revealed higher yields from codon-optimized variants (Figure 4). Exemplifying this trend, the relative soluble abundance of the rRNA biogenesis protein RRP5 (HTP4400) was found nearly three times higher when expressed from its codon-optimized variant compared to its harmonized equivalent (Figure 4); corresponding to a yield difference of ~110 mg/L (Table 1). The codon-optimized gene variant of hypothetical protein (HTP4360) was also expressed at a high level, with an estimated yield of ~150 mg/L soluble protein. Endopeptidase tail protein (HTP4435), amidase (HTP4410) and DUF262/DNase (HTP4350) expressed from codon-optimized gene sequences demonstrated only slightly higher relative abundance (by 2–5%, respectively,) compared to respective codon-harmonized variants (Figure 4). The soluble yields of HTP4435, HTP4410 and HTP4350 from codon-optimized variants were also found to be ~8–16 mg/L higher than the yields of the corresponding codon-harmonized variant (Table 1). Following this step, target proteins from both variants, which were noted as soluble, were up-scaled to be produced in 1 L expression cultures.

### 3.4. Protein Purification

Soluble proteins produced from 1 L cultures were purified to near homogeneity by nickel affinity chromatography. An optimized affinity chromatography purification protocol ensured high purity of the target proteins as was visualized by SDS-PAGE (Figure 5), where target proteins were observed at bands corresponding to their expected sizes. Purified endopeptidase tail protein (HTP4435), expressed from both types of codon-adjusted gene variants, were aggregation-prone, while the other target HTH1 proteins remained stably soluble after purification. The single step purification strategy led to generally high purification yields (Table 2). Comparison of the obtained yields of amidase (HTP4410) as well as DUF262/DNase (HTP4350) expressed from codon-optimized and codon-harmonized gene sequences did not differ, whereas the purification yield of codon-harmonized rRNA biogenesis protein RRP5 (HTP4400) was approximately 20% higher compared with the yield of its codon-optimized gene counterpart. In general, the purification yields confirmed a comparatively high affinity of heterologous proteins towards the chromatography resin and were in the expected range for the method [94,95].

### 3.5. Crystallization and Thermostability of Target Proteins

Purified, stably soluble target HTH1 proteins expressed from the optimized and harmonized types of codon-adjusted gene variants were subjected to both crystallization trials and analysis of thermostability. As a higher thermal unfolding temperature has been indirectly connected to an improved fold, that may affect the possibility to crystallize the target protein. In crystallization trials, amidase (HTP4410) as well as DUF262/DNase (HTP4350) expressed from codon-harmonized gene variants (Appendix A) and rRNA biogenesis protein RRP5 (HTP4400) from both codon sequence adjustment variants were observed to form protein crystals (M. Håkansson and S. Al-Karadaghi, SARomics Biostructures, personal communication).

In the thermostability assessment with differential scanning fluorometry, which was performed to compare melting temperatures (T_m_) of target recombinant proteins expressed from both types of codon-adjusted gene sequence variants, an increase in unfolding temperature was observed from the codon-harmonized variants of the three target proteins where crystal formation was observed. The in vitro thermostability (T_m_) of the target HTH1 proteins amidase (HTP4410), rRNA biogenesis protein RRP5 (HTP4400) and DUF262/DNase (HTP4350) varied between approximately 51 and 73 °C. Remarkably, recombinant proteins expressed from the codon-harmonized gene variants were all observed to unfold at higher T_m_ values (3–7 °C) than corresponding codon-optimized gene variants (Table 3). A T_m_ of approximately 61 °C was determined for DUF262/DNase (HTP4350) expressed from a codon-optimized gene variant, which was an almost 3 °C lower unfolding temperature compared with the T_m_ observed for this hypothetical protein expressed from the codon-harmonized version. Amidase (HTP4410) and rRNA biogenesis protein RRP5 (HTP4400) expressed from codon-harmonized gene sequence versions demonstrated a T_m_ at 73 °C and 56 °C, respectively—increases of almost 7 and 5 °C compared to the T_m_ of proteins expressed from codon-optimized genes.

## 4. Discussion

Marine bacteriophages remain a largely unexplored resource for enzyme bioprospecting. As a part of the Virus-X consortium (http://virus-x.eu/, accessed on 1 May 2021), successful expression of genes from bacteriophage genomes was identified as a key step towards discovering enzymes from various marine niches. Crystallization of novel viral proteins to collect structural knowledge was another aim of the consortium, as recently exemplified for the proteins XepA and YomS from a *Bacillus subtilis* prophage [96]. Hence, significant research interest currently exists for the analysis of new phage genes that may hold interest both in basic and structural research and for applications in biotechnology.

In this context, a novel prophage, designated HTH1, was identified via the study of the Gram-negative hydrothermal vent bacterium *H. thermotrophus*, which is classified in the phylum *Fusobacteria*. The relationship between *H. thermotrophus* and HTH1 can be considered fitting, as lysogeny is suggested to be prevalent in physiochemically demanding environments. These include deep-sea biomes [97] and diffuse-flow hydrothermal vent communities [22], where temperate phages may provide benefits to host fitness via various mechanisms [98,99,100].

Taxonomic analyses placed HTH1 within the family *Siphoviridae,* which contains dsDNA viruses defined by their long, non-contractile tails, as opposed to the contractile tails of the *Myoviridae* and the short and non-contractile tails of the *Podoviridae* [101]. The genome size of HTH1 was 41571 bp, indicating it to be smaller compared to the average genome size of *Siphoviridae* at ~53 kb [102]. Interestingly, phylogeny (Figure 1), Virfam [66] and sequence homology analyses of HTH1 genes (Appendix A) all suggested closest similarity of HTH1 to siphoviruses that infect Gram-positive bacteria, mainly of the phylum *Firmicutes.*

HTH1 was annotated to contain a suite of expected viral backbone genes, such as structural elements for the viral head, neck, capsid and tail, core viral enzymes such as integrases, terminases, the viral lytic enzymes, and DNA modifying enzymes such as DNA polymerase, endonuclease and recombinases (Appendix A, Figure 3. However, further studies including the lytic induction and isolation of viral particles would be required to confidently determine whether the presented genome of HTH1 corresponds to the complete and functional phage genome infecting *H. thermotrophus*.

Closer inspection of the HTH1 lytic cassette revealed three main genes related to cell lysis: a glycosyl hydrolase putatively capable of chitin and peptidoglycan-degrading activities specific to endo-β-N-acetylglucosamine residues [103,104]; a holin crucial for the perforation of the cell membrane [105,106]; and an N-acetylmuramoyl-L-alanine amidase featuring a membrane binding lysin motif (LysM), with an expected activity of cleaving bonds between N-acetylmuramoyl residues and L-amino acids in the bacterial cell wall (Figure 2 and Figure 3). However, no genes related to spanins, rod-like viral lysis proteins considered essential to disrupt the cell membranes of Gram-negative hosts, were detected [106,107].

The enzymes of the HTH1 lytic cassette, containing the genes annotated to encode glycosyl hydrolase, holin and amidase, were of obvious interest as their peptidoglycan-degrading capabilities could be utilized against pathogenic bacteria as bactericidal agents [108]. In addition, the hypothetical protein HTP4435 was selected for testing due to the presence of a tail-associated endopeptidase domain (Pfam PF06605, MEROPS M23) (Figure 3). Such peptidases may find a broad range of potential uses in industrial, medical or scientific applications [109,110,111]. The DNA polymerase I (HTP4385) was also of direct interest for its potential as an enzymatic tool in many modern molecular biological methods such as PCR, genome sequencing and more [112]. As *H.*
*thermotrophus* was reported to grow optimally at 48 °C [55], the proteins encoded by HTH1 may possess elevated thermostability and thermal activity, which are desirable traits in many industrial or scientific applications [113,114]. Furthermore, only limited structural similarity was observed for the chosen HTH1 proteins to structures present in PDB (Appendix A), suggesting novel features could potentially be revealed with their future structural analyses.

The heterologous expression of native phage proteins has been reported to be challenging [115]. To aid in this process, codon optimization [49] and codon harmonization [50] approaches were considered for the heterologous production of proteins encoded by HTH1. Here, these two approaches were tested, and compared over their effects towards obtaining and increasing soluble protein yields, and also for their effects on the thermostability of the proteins produced. While codon optimization is commercially offered as an option during gene-synthesis services [116], codon harmonization must be carried out manually, and so a deeper understanding of the native viral host is required. As bacteriophages can naturally use their host’s machinery to express their genes, they are understood to adapt the same codon usage frequency (CUF) as the host [117]. Therefore, while preparing sequences for codon harmonization, the genome of *H. thermotrophus* was used to calculate and compare CUFs between itself and *E. coli* as the expression host.

Codon analysis of selected native HTH1 genes suggested the target proteins are naturally produced in moderate amounts in *H*. *thermotrophus*. As expected, heterologous target proteins were produced more readily from codon-optimized gene variants than comparable codon-harmonized genes (Table 2, Figure 4 and Appendix A), which were adjusted to mimic the gene native codon landscape, sacrificing overall codon adaptation to the expression host in the process [50]. The codon optimization approach for selected HTH1 proteins was successful, as quantitatively confirmed by estimated CAI values and also by observed soluble expression yields. The CAI for the codon-harmonized variants of selected genes were comparatively high and varied substantially, indicating that the codon harmonization algorithms used [83,118] were suitable and specific for each of the HTH1 genes.

Protein folding quality is typically reflected by a higher thermal unfolding temperature and a higher thermostability [119]. While the codon harmonization approach did not result in the soluble expression of a greater variety of HTH1 proteins than codon optimization, it yielded proteins with comparatively higher melting temperatures (T_m_) determined by differential scanning fluorimetry, suggesting a higher folding quality. Assayed under identical conditions, higher unfolding temperatures were observed for all HTH1 target proteins expressed from codon-harmonized gene variants compared to corresponding proteins from codon-optimized variants. The melting temperatures determined were in an expected range for HTH1 proteins natively produced within the host cells, fitting with the optimal growth temperature of *H. thermotrophus* [55]. Furthermore, ongoing crystallization trials also confirmed better crystal-forming properties of target HTH1 proteins expressed from codon-harmonized genes as an indicator of improved folding quality (M. Håkansson and S. Al-Karadaghi, SARomics Biostructures, personal communication).

The CHI values estimated for codon-harmonized variants of the selected gene set were comparatively high and did not differ substantially between the different genes in the set, indicating moderate, if not limited harmonization of codons (Table 1). These results could partially explain why target proteins produced from codon-harmonized variants were not persistently more soluble than codon-optimized variants after production in *E*. *coli*. In theory, production of soluble proteins should be ensured by codon harmonization [84], even though further optimization of physiochemical heterologous expression parameters is recommended to enhance the expression level of soluble protein from codon-harmonized gene variants [120]. Preliminary experiments to express selected HTH1 genes in *E*. *coli* were carried out under the recommended conditions for the expression vector and strain used [85]. Further optimization of the process could be implemented to achieve soluble production of target proteins, which remained insoluble despite codon harmonization. As the current codon adjustment algorithm was mainly developed using non-viral genome sequences, its efficacy could be limited for the adjustment of viral genes. With the limited data available for the implementation of codon adjustment for viral genes [121,122], the results presented herein may aid the further development of codon adjustment algorithms.

## 5. Conclusions

In this work, complementary application of bioinformatics and molecular methods allowed the identification, description and protein-level study of a novel marine prophage. Here, we describe the first genome sequence of a prophage discovered in *H. thermotrophus*, a Gram-negative, moderately thermophilic bacterium isolated from the Seven Sisters hydrothermal vent field. The *H. thermotrophus* phage H1 (HTH1) showed similarity to phages infecting Gram-positive bacteria of the genus *Firmicutes*, but in our study, it was found within the genome of a Gram-negative host. A set of nine genes were identified with putative functions, including cell lysis, nucleotide lysis and replication—interesting for both ecological studies and potential biotechnology applications. To facilitate the soluble heterologous production of HTH1 proteins in *E. coli*, codon optimization, and harmonization approaches were tested in parallel. Valuable data regarding production yield, solubility and folding quality of heterologous HTH1 proteins were gathered following expression of codon-adjusted gene variants, which may be useful in improving the application of codon adjustment strategies for viral genes. In the context of the proteins tested, codon optimization was found to lead to higher protein yields, whereas codon harmonization was underlined as more beneficial for the production of proteins with higher stability and folding quality.

## Figures and Tables

**Figure 1 viruses-13-01215-f001:**
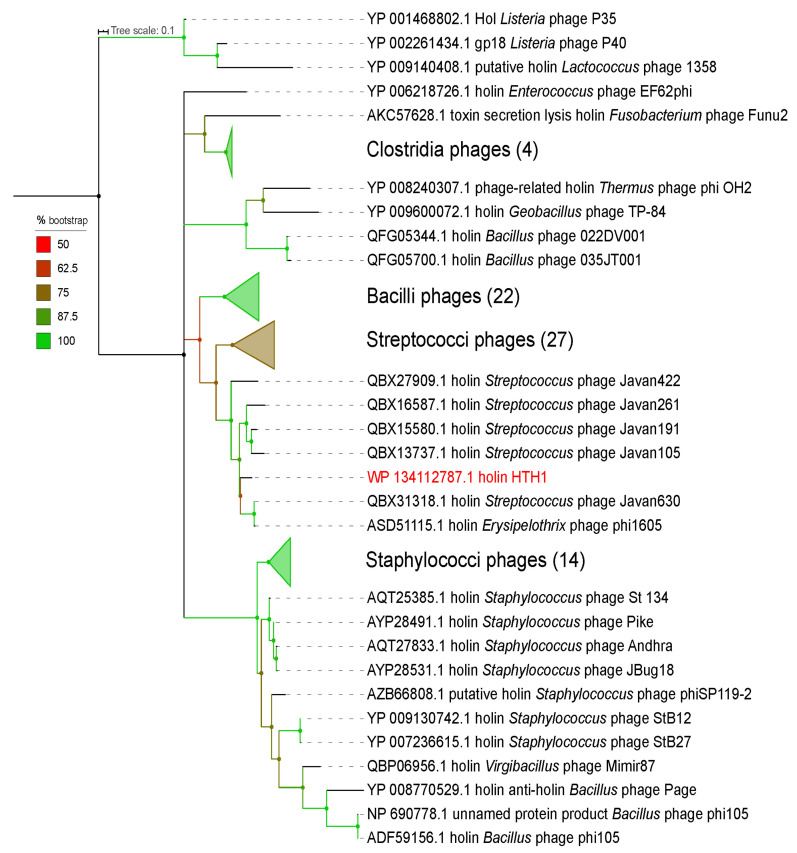
Phylogeny analysis of the prophage based on the alignment of 106 amino acid long region of holin proteins from 94 phages, using maximum likelihood, with 1000 bootstrap replicates. The tree is centre-rooted, and the scale bar represents the average number of amino acid substitutions per site. Numbers next to collapsed clades represent the number of leaves covered by each illustration. The HTH1 holin is highlighted in red.

**Figure 2 viruses-13-01215-f002:**
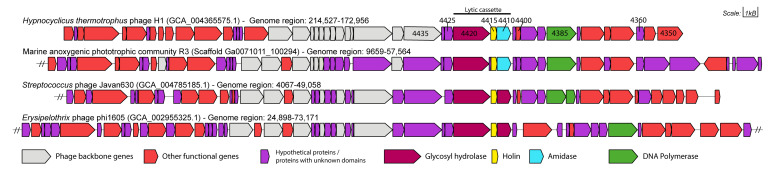
Gene neighbourhood map of HTH1 and the comparable regions of three closely related phage gene clusters aligned around the holin in their respective lytic cassettes. Displayed genes are drawn to scale, as shown on the top right. Respective organism or sample names, related accession numbers (in parentheses) and genome regions displayed (in bp ranges) are provided above each graphic. Genes chosen for expression of proteins from HTH1 are also labelled with their identifier numbers. Double dashes (//) indicate the presence of genes further up or downstream the gene regions displayed in this figure.

**Figure 3 viruses-13-01215-f003:**
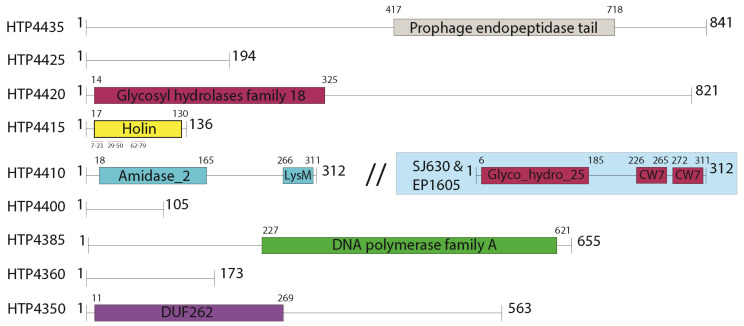
Illustration depicting sequence features of chosen candidate proteins predicted by HMMER [82]. Black lines show non-annotated amino acid sequences, grey boxes show predicted Pfam domains, purple lines mark transmembrane domains and numbers flanking each feature show their respective amino acid residue number ranges. The blue box shows the HTP4410 analogue found in *Streptococcus* phage Javan630 (SJ630) and *Erysipelothrix* phage phi1605 (EP1605).

**Figure 4 viruses-13-01215-f004:**
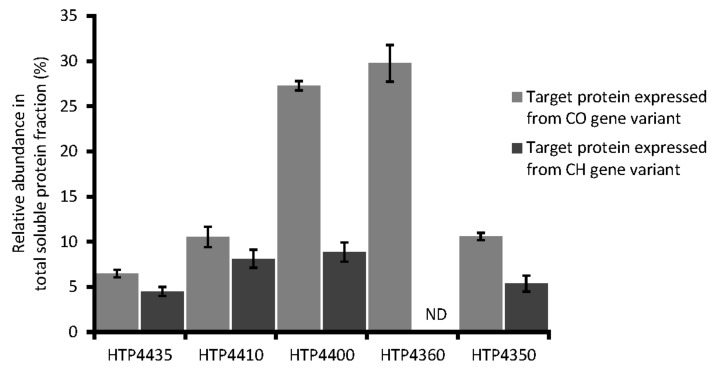
Relative abundance of target HTH1 proteins produced after expression from codon-optimized (CO) and codon-harmonized (CH) gene variants in total soluble protein fraction. ND—target protein not detected in total soluble protein fraction. Values represent relative abundance mean in percent of total proteins in total soluble protein fraction ± standard error of three independent expressions.

**Figure 5 viruses-13-01215-f005:**
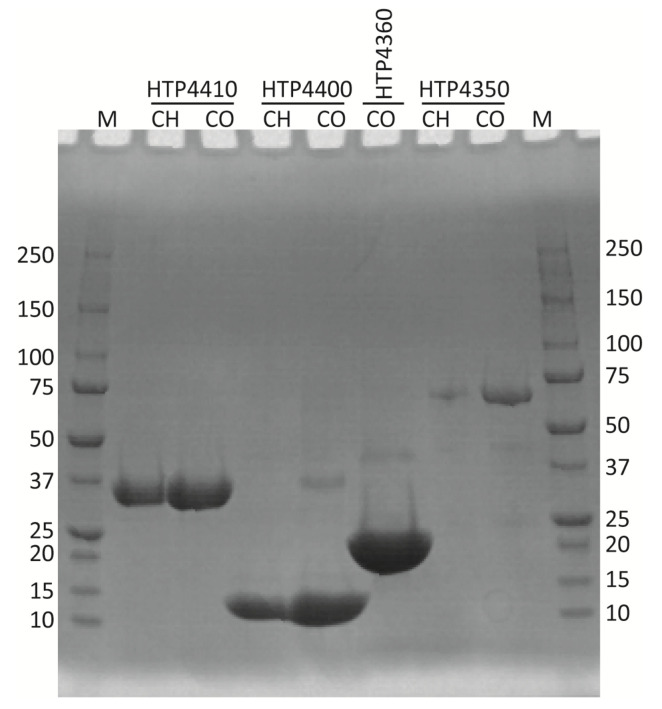
SDS-PAGE image of purified proteins produced from codon-harmonized (CH) and codon-optimized (CO) genes. The HTP prefix and the numbers above the lanes correspond to the identifiers of the genes tested. M indicates the protein marker (Bio-Rad Precision Plus Dual Color). Numbers next to each protein marker lane show the respective molecular weight labels in kDa.

**Table 1 viruses-13-01215-t001:** Codon usage parameters and soluble production yield estimation of target HTH1 proteins. CAI—codon adaptation index, CHI—codon harmonization index, CO—codon-optimized, CH—codon-harmonized, ND—target protein not detected in total soluble protein fraction.

Identifier	Proposed Protein Function	CAI for Expression Host		CHI for Expression Host		Codon Native Gene Sequence CAI for Native Host	Soluble Produced Protein Yield * (mg/L)	
		CO Gene Variant	CH Gene Variant	CO Gene Variant	CH Gene Variant		Expressed from CO Gene Variant	Expressed from CH Gene Variant
HTP4435	Endopeptidase tail	0.89	0.64	0.61	0.48	0.50	18.7 ± 1.3	13.8 ± 1.5
HTP4425	Hypothetical protein	0.87	0.67	0.59	0.48	0.51	ND	ND
HTP4420	Glycosyl hydrolase 18	0.89	0.66	0.60	0.45	0.51	ND	ND
HTP4415	Holin, toxin secretion/phage lysis	0.87	0.58	0.60	0.47	0.40	ND	ND
HTP4410	N-acetylmuramoyl-L-alanine amidase	0.88	0.64	0.61	0.47	0.46	40.7 ± 5	30.8 ± 3.4
HTP4400	rRNA biogenesis protein rrp5, putative	0.84	0.64	0.58	0.48	0.48	135.80 ± 2.49	27.9 ± 3.2
HTP4385	DNA Polymerase	0.86	0.62	0.60	0.46	0.47	ND	ND
HTP4360	hypothetical protein	0.84	0.74	0.55	0.43	0.56	151.7 ± 10.2	ND
HTP4350	DUF262 / DNase	0.86	0.62	0.59	0.45	0.44	32.3 ± 1.2	15.6 ± 2.7

* Values represent mean ± standard error of three independent expressions.

**Table 2 viruses-13-01215-t002:** Purification yield of target HTH1 proteins. Protein concentrations were measured spectrophotometrically estimating total amount of target recombinant protein in clarified lysate by combining densitometry calculation results and total soluble protein quantification results. CO—codon-optimized, CH—codon-harmonized, ND—target protein not detected in total soluble protein fraction.

Identifier	Proposed Protein Function	Protein Purification Yield * (%)	
		Target Protein Expressed from CO Gene Variant	Target Protein Expressed from CH Gene Variant
HTP4410	N-acetylmuramoyl-L-alanine amidase	85.6 ± 1.4	85.9 ± 1.9
HTP4400	rRNA biogenesis protein rrp5, putative	38.6 ± 7.2	58 ± 3.7
HTP4360	hypothetical protein	75 ± 3.8	ND
HTP4350	DUF262/DNase	83.5 ± 5.2	92.1 ± 2.1

* Values represent mean ± standard error of three independent purifications.

**Table 3 viruses-13-01215-t003:** Thermal unfolding estimation with differential scanning fluorimetry of stably soluble target HTH1 proteins. CO—codon-optimized, CH—codon-harmonized.

Target Protein	Proposed Protein Function	Melting Temperature (T_m_, °C)	
		Target Protein Expressed from CO Gene Variant	Target Protein Expressed from CH Gene Variant
HTP4410	N-acetylmuramoyl-L-alanine amidase	66.23 ± 0.07 *	73.03 ± 0.10
HTP4400	rRNA biogenesis protein rrp5, putative	51.57 ± 0.34	55.70 ± 0.22
HTP4350	DUF262 / DNase	61.57 ± 1.47	65.24 ± 0.43

* Values represent mean ± standard error of three independent differential scanning fluorimetry assays.

## Data Availability

All relevant data for the study is provided within the article, and its supplements.

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
