# Peer review of "Exploring Codon Adjustment Strategies towards Escherichia coli-Based Production of Viral Proteins Encoded by HTH1, a Novel Prophage of the Marine Bacterium Hypnocyclicus thermotrophus"

_viruses, 2021, doi:10.3390/v13071215_

Round 1
Reviewer 1 Report
The authors did their very best to improve the paper with an obvious success. A few small editorial details request to be solved:
-Figure to is cut on its right.
- In all tables, the results are given with four digits which is nonsense. Three or enven 2 digits woulde be sufficient, especially concerning the purification yields
Author Response
The authors appreciate the reviewer's feedback. Figure 2 has been resized to better fit the page bounds. In addition, values in the tables have been adjusted to depict purification yields with only one value after the decimal point. The values for Tm measurements in Table 3 were considered significant for this experiment, and were therefore left as before.
Reviewer 2 Report
The manuscript “Exploring codon adjustment strategies toward Escherichia coli-based production of viral proteins encoded by HTH1, a novel prophage of the marine bacterium Hypnocyclicus thermotrophus ,” by Arsin et al., describes using bioinformatic techniques to identify a novel temperate phage in the genome of the host. This is interesting in itself, as the analysis of phage functions from Smoker Vent phages compared with phage from different environments could be a very fruitful study. However, following some initial characterization of the new phage, which included functional annotation but NOT phage isolation, the authors switch gears to discuss the potential usefulness of enzymes derived from very hot environments. They proceed to express a specific subset of phage genes in E. coli, resulting in the interesting discovery that while codon optimization results in a higher protein yield, codon harmonization yields proteins with greater stability and better crystallization properties, implying that CH results in improved protein folding.
I found this paper interesting to read, and the data seem thorough and convincing. My main problem is that the focus is unclear and confusing – is the focus novel phages from interesting environments? or gene expression and protein folding in E. coli? The transition from one topic to the other should be more seamless and make logical sense. Right now it’s jarring.
For example, the authors go to great lengths to characterize the phage, including inferring that it’s a siphovirus. The extensive characterization implies a focus on the phage itself, not just as a bank of genes to furnish tools for biotechnology. Yet there’s no mention of phage growth. I believe that this host can be grown anaerobically in the lab – was there any attempt to isolate phage from the supernatant and try to plaque it or identify it using qPCR? It seems odd that this type of experiment is not included, unless the major focus is really gene/protein expression. If so, then the extensive phage characterization should still be reported, but mainly through figures and tables. It should be de-emphasized in the text.
In summary, I feel the ms should be published, but unfortunately, it needs further rewriting to smooth over the disparate parts and to clarify the focus.
Author Response
The authors appreciate the detailed response from the reviewer. The study aimed to provide valuable data on the effect of codon adjustment approaches on heterologous expression of viral genes, explored over an interesting research target - in this instance the novel prophage found within H. thermotrophus. The authors agree that the manuscript would read smoother without the deep dive into comparative analysis of genes of discussed phages. As advised by the reviewer, the manuscript has been revised to de-emphasize the deep written description of gene similarities, but relevant figures and supplementary data are still provided for the interested reader.
Round 2
Reviewer 2 Report
The manuscript is definitely improved over the prior version, as much of the "deep dive" into the phage genes has been removed. At this point I would say that two things remain to be addressed:
- It should be made clear that it is not known whether the newly discovered prophage is a complete and functional phage genome.
- The Discussion section is very verbose, and would benefit from careful editing of overexplained concepts. I would also remove, or greatly abbreviate, the section from lines 620 to the end, as it doesn't really add anything.
Author Response
The authors appreciate the reviewer's continued feedback. The manuscript has been further edited in the suggested direction.
- A statement regarding this matter has been included in line 500 to reflect our knowledge level on genome completeness.
- The discussion section has been further edited for brevity.
This manuscript is a resubmission of an earlier submission. The following is a list of the peer review reports and author responses from that submission.
Round 1
Reviewer 1 Report
The manuscript by Arsin et al. describes in a first part the genome identification and analysis of the prophage HTH1 present in the genome of the hydrothermal vent bacterium Hypnocyclicus thermotrophus. In a second part, the authors evaluated the biotechnological potential of 9 genes selected on the premise that they could be involved in cell lysis and DNA replication.
In the first part, that describes and analyses the genome of the phage, the authors state that the phage belongs to the Siphoviridae family, based on the taxonomic analysis of two regions, and that of head-tail joining modules belonging to a type strictly present in siphophages, but only in those infecting the gram-positive Firmicutes. Here, we have a gram-negative bacterium, so the argument is weak. A strongest argument would be to examine the morphology of the phage by EM, but I guess that this bacterium cultivation is a tough task.
Since siphophages possess a long non-contractile tail, examination of the tail tape measure protein (TMP) would allow to discard a Podoviridae. The absence of a tail sheath protein would then discard the Myoviridae hypothesis. Another clue is the size of the genome (not provided here) that is smaller in Siphoviridae.
A weak point of the presented work is therefore the annotation. The TMP, one of the easiest proteins to annotate in siphophages, due to its position and its length is not identified here. Tail proteins at the end of the structural cassette are also not identified (only as tail proteins and hypothetical proteins). HHpred (1), a very powerful tool, makes it possible to annotate a protein based on its secondary structure and identities in the twilight zone (10-20%) It could have been used here.
The most problematic part, however, is that dealing with the 9 genes expression. Since the years 2000-2010 and the development of structural genomics, we know that expression of soluble proteins should be assayed using and varying different parameters: i/constructs, including fusion with expression helpers (MBP, thioredoxin, GST), ii/ different temperatures, including temperatures between 15 and 20C, iii/ culture medium, iv/ coli strain, etc (2). Here the authors use one construct, one temperature (20C) and one culture medium. While codon-optimization will deal with the overall production yield, including the inclusion bodies, they will not play a role in getting the protein soluble. Amazingly, the authors have included holin, a membrane protein, in the assay, while it can be solubilized only in the presence of detergents.
Finally, the analysis of solubility is also superficial, as it involves only the comparison of the SDS gels of the total protein and the clear lysate (no centrifugation!). They do not even use the His tag present in pET-21b to analyse and separate the protein by affinity. Mass spectrometry of the putative expressed proteins in the gels would have been also mandatory to confirm their identityIn my opinion, this part of the work is not conclusive.
(1) HHpred . https://toolkit.tuebingen.mpg.de/tools/hhpred
(2) Protein production and purification. Graslund, S et al. NATURE METHODS, 5, 135-146 (2008) DOI: 10.1038/nmeth.f.202
Reviewer 2 Report
This is a nice study that has identified and cloned a number of proteins from an organism isolated from deep sea vents. The genes encoding this proteins have been altered to imporve expression in E. coli. The potential for these to proteins to be of biotechnological importance is there, but this has not been realised and their activities have not been proven.
The biggest issue in this manuscript is the unusual manner in which this group has chosen to make this paper focus on phages, which feels like a means of justifying submission to this journal.
The assigning of this prophage or ctyptic prophage sequence into the Siphoviridae is over laboured. The grouping was originally based on virion morphology and requires the presence of a long flexible tail. No data has been provided on the tail. The capsid and neckstructures do not ensure that the virus has a long flexible tail. Phage sequence evolution results from two main drivers......mutation within a host and recombination of DNA seqeucnes within a host (host with phage, phage with phage and phage with other mobiel genetic elements). Trying to force phage phylogeny with polymerases and other enzymes that would have been picked up anywhere makes very little sense. There is also the issue of host lysis genes related to Gram positive organisms in a Gram negative host. This makes little biological sense, unless biology and physiology bacteria is greatly altered in the deep sea. Just because the genes are there....especially in a prophage or prophage remnant doesn't mean they are biologically active if Horizontal gene transfer is active.
I would focus this study back onto the enzymes you have cloned and their activity.